# Rethinking Legal Criteria for Assessing Compensation for Rural Land Expropriation: Towards a European Institutional Framework

**Anastasia Hernández-Alemán [1], Noelia Cruz-Pérez [2] and Juan C. Santamarta [2,*]**

1    Instituto de Turismo y Desarrollo Económico Sostenible (TiDeS), University of Las Palmas de Gran Canaria, 35001 Las Palmas de Gran Canaria, Spain; anastasia.hernandez@ulpgc.es
2    Departamento de Ingeniería Agraria y del Medio Natural, Universidad de La Laguna (ULL), 38200 Tenerife, Spain; ncruzper@ull.edu.es
*    Correspondence: jcsanta@ull.es

**Abstract:** In public management, it is common to face conflicting objectives, particularly in relation to land use. Adequate land use management requires a valuation of land that incorporates the value of all its characteristics. That is, in addition to the traditional direct use value, it must incorporate the non-use value (existence and legacy), as well as the indirect use and option values. The analytic hierarchy process is used, firstly, to identify the priority values based on a panel of experts, and secondly, in assessment of use/non-use values, using market valuation techniques as support. As a result, we analyse the trade-offs among all values, and the respondent's consistency. At first, we observed that the soil with the highest protection had the lowest market value in terms of direct use. However, considering the weights of the panel of experts, we can conclude that the market value only represents 7.6% of the total value. Non-market values represent 92.4% of the total value. The underlying aim is to facilitate decision-making in the field of land management to increase social welfare and the resilience of landscapes.

**Keywords:** natural park management; analytic hierarchy process; economic valuation

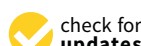



## 1. Introduction

This study provides a contribution in relation to governance challenges and strategies for reconciling landscape resilience with trade and development. One of the major issues in the literature refers to the environmental services assessment, which involves how to reconcile environmental services and socio-economic needs, especially from people who live in a protected natural area. Protected areas as an applied tool for biodiversity conservation, environmental goods and services which cannot be traded explicitly in the market because of social welfare, specific methods of valuation, and institutional frameworks should be considered jointly to achieve environmental sustainability [1–6]. Traditional methods used to assess the soil as urban or rural only take into account socioeconomic factors, or the market value of goods and services obtained from the productive process, without considering the value of the positive externalities generated by the natural ecosystems associated with the soil being valued. Environmental valuation allows us to assign monetary values to different public interventions, to provide support for interventions of policy-makers considering environmental damage, valuing positive externalities, and the complexity of the area [7–19]. In Spain, this type of valuation is regulated by Royal Decree 1492/2011 of 24 October, which approved the valuation regulations of Land Law, and by Royal Decree 7/2015 of 30 October. As a result of the practice established in the regulations, asymmetry is generated in the information obtained from the valuation, which is transferred to public decision-making [20–24]. Soil produces market and non-market goods and services. The question that arises, and which is the subject of analysis in this

paper in a specific case study, is to analyze how it would affect the value of land taken as a reference for public decision-making if, in addition to direct use values (traditional methods), non-use values (existence and legacy), option values and indirect use values were considered. We find in the literature numerous examples of valuation that address the complex problem of valuing goods and services that are lacking an explicit market for their exchange [1,25–38]. Greiner et al. [33] reviewed literature that showed the lack of ecosystem services (ES) and evaluation studies that take into account the multifunctionalities of soils, providing methods approved by the German Federal States in spatial planning procedures. In the field of economic valuation of ecosystem services (ES), Paletto et al. [36] tested the economic value (Figure A1) of ES obtained from different forest management practices, concluding that physical and economic integration in ES valuation is essential to evaluate trade-offs and provide multiple perspectives for forest policy formulation. This paper uses as a case study the Natural Park of Tamadaba in Gran Canaria (Canary Islands, Spain), whose declaration as a protected area is based primarily on conservation of the forest mass of the Tamadaba massif. Mountain forests are characterized by multiple functions that must be adequately valued for management purposes [13,19,39–48]. The reflection is supported in a specific case study, as we can see in the previous literature, using qualitative methods [1,19,30,34,35,42,45,46]. However, the aim of this work is not so much to illustrate a specific case study, but to make evident the need for an institutional framework that could be a support for land valuation particularly for rural or environmental uses [33,44]. The reflection involves governance in a wide sense (policies, institutions, process of decision-making, valuation, etc.). European institutional framework provides a wide range of legal tools to protect biodiversity [2,4]; however, there is a lack of these relating to the valuation of rural or environmental land. Many few European projects need land for their development, some of which enter into conflict with those projects for sustainable use. This is why this reflection should be transferred to a broader legal framework than Spain's, although it serves as an example [5,19,49].

The methodological constraint provided for in the regulation is based on the need to avoid land speculation, which undoubtedly comes into play when individual, business and social expectations are taken into account. However, the opposite happens in practice and, far from correcting speculative movements in the exchange of land, inadequate valuation in the field of public management leads, on most occasions, to relocation of public and private infrastructure to where the land is more economical, which happens to be land for agricultural, environmental or recreational purposes. In practice, their protection is disaffected, reclassifying the land, or expanding its uses in urban planning. Budgetary restrictions therefore seek to ensure that compensation or land acquisition is as small as possible. Otherwise, this would lead us to rethink the investment project and its possible alternatives, or even to discard it. Tgalarino et al. [49] analyzes in relation to "compensation for expropriation", non-compliance with the International Standards on the Valuation of Compensation (ISVC). We cannot say that Spain does not comply with international valuation standards for the purpose of compensation for expropriation, as this is even provided for in the Spanish Constitution, but we can say that, in practice, the environmental aspects, the value of ecosystems and the social welfare they provide, are notably neglected, and this leads to a distortion or asymmetry in the valuation process [19,50–58]. Without taking into account the judicialization of expropriation processes due to disagreements in the valuation, which is very frequent in practice, in most cases this means receiving the monetary amount in terms of compensation more than 10 years after occupation of the land, which is far from the principle of justice. The damage to the value of rural land through institutional channels is increased if we take into account the Supreme Court ruling of 8 June 2020 (ref. BOE-A-2020-14654), abolishing the corrective coefficients provided for in the law which, in practice, reflect the risk associated with the type of crop. However, these corrective coefficients are maintained in relation to extractive, commercial, industrial and service activities. The situation of unprotected environmental land where production is not possible is even worse, since the rent that is updated to obtain its value is one third of

the theoretical rent (rent for productive land, or rental fee in practice). Notwithstanding the foregoing, the regulation gives a nod to the environmental value of the land (art. 7.3 and art. 17.5 of the Royal Decree 1492/2011). Thus, when the rural land to be valued is located in areas of singular environmental or landscape value, the corrective factor $u_3$ will be applicable, defined as: $u_3 = 1.1 + 0.1 \, (p + t)$, where $p$ is the weighting coefficient according to the environmental or landscape quality. The weighting coefficient "$p$" must be determined on the basis of objective criteria in accordance with the values recognized for the land under appraisal in urban and territorial planning instruments or, as the case may be, in networks of protected spaces. It will be between 0 and 2, and it will take into account the values and qualities of the surroundings. The higher the number, the better the environmental and landscape quality or cultural, historical, archaeological and scientific aspects. It is up to the appraiser to justify the weighting coefficient, since the legal text does not provide any method or rule for its calculation and, in this sense, the reflection of Bartke and Schwarze [24], in reference to the role of the appraiser and the information asymmetry, information acquisition, information brokerage and expert knowledge is appropriate. The legal text establishes that the weighting coefficient "$t$" can reach a maximum value of 7 when the land and urban planning instruments allow a regime of uses and activities other than agricultural or forestry uses and activities that increase the value. The methodology and the criteria to be followed in land valuation practice in the field of public and private management lead to a clear predominance of direct use values that, in most cases, generate strong speculative movements of land to the detriment of social welfare. The question posed in this paper is: what would happen in public and private decision-making if this "asymmetry" in the valuation of rural land were corrected by incorporating all of its environmental values? We illustrate the circumstance described above and, thus, we can see that the m$^2$ of rural land in the exclusion zone (scientific and conservation use) is worth 2.99 €/m$^2$ according to the valuation rules of the standard. The m$^2$ in the special use zone, which allows certain uses typical of urban land, has a value of 264.43 €/m$^2$. In order to support this, it is simple and convenient to use a multi-criteria decision method (MCDM) to solve decision-making problems based on multiple criteria [9,11,12,59–72]. The choice between at least two alternatives (carrying out an investment project or not) requires an adequate assessment of the benefits and costs of choosing the alternative that maximizes social welfare [73]. An appraisal is required that articulates each of the characteristics of the asset being appraised [8,63–65,71]. In this way, it is society that chooses the alternative that optimizes its welfare, through its representatives in the framework of public decision-making. It seems clear that the existence of a natural environment whose maximum recognition is given to a form of protection, be it a nature park, a rural park, a protected landscape, or a site of scientific interest, among others, brings advantages and disadvantages, particularly from an economic point of view [74]. However, there is a debate in the literature regarding the role of institutions and the most appropriate methodology for structuring values that reflect social preferences. It is still a challenge, both from a methodological point of view and the management of natural spaces, particularly in terms of forest management and the valuation of the environmental and socio-economic services that these spaces provide to the local resident communities and to visitors [75]. The purpose of valuation is not to assign a price to a natural area that could serve as a reference for commercial exchange. Many of the services provided by the natural area, particularly with regard to ecosystem services, do not have their own market due to the characteristics of these services [18,68,76]. The main problem we encounter in carrying out the valuation of environmental services or the externalities generated by environmental assets has to do with the characteristics of public goods, such as non-exclusion (when the good or service is offered, no one can be excluded from enjoying it even if they do not pay for it); and non-rivalry in consumption (when someone consumes the good or service, it does not reduce the consumption of other potential users). Public goods and common resources, which would be those whose property rights are not defined [20], are a particular case in the framework of externalities. We may consider an externality to be

the activity of a person, whether natural or legal, which affects the well-being of another without being able to charge a price for it, whether in a positive or negative sense. In this sense, we could say that there is a market failure because there is no payment for the negative externalities, and neither is there an economic benefit for positive externalities. Therefore, in the absence of an adequate definition of property rights, together with market inefficiency, the intervention of the state is required to establish an adequate allocation of public resources [58]. However, the fact that markets cannot assign a price to certain goods and services and are, therefore, not interchangeable does not mean they are worthless. Ronald Coase, who received a Nobel Prize in Economics in 1991, solved the problem of the absence of a market for externalities by establishing that, in the absence of transaction costs (operational costs), the problem caused by externalities could be solved by assigning the property rights to one of the parties [23]. The so-called Coase Theorem [21] not only implies that all parties affected by the externality are known, but also that negotiations are possible in order to reach a price. A limitation to Coase's contributions is that not all agents have the same influence or bargaining power, so it is very likely that large organizations will have greater capacity to influence agents suffering from externality, which would undoubtedly lead to environmentally questionable results.

The novelty of this paper and approach consists of providing an alternative methodology that allows management decisions to be taken in relation to land use, articulating all its possible values, taking into account its environmental characteristics. This is in addition to providing a reflection on the restriction of land valuation by reason of compensation in Spanish legislation and its harmful effects, as described in the previous section. The next section describes the study area for which a valuation is made on the basis of direct use values. The following sections describe the theoretical framework and the methodology used for the total valuation, which includes qualitative and quantitative characteristics of the study area. The obtained results are analyzed. The final section offers concluding remarks. For a better understanding of the different types of values considered in the analysis, the survey used for the case study describes each one in Appendix A.

## 2. Study Area

Tamadaba Natural Park, on the island of Gran Canaria, Spain, has an area of 7538.6 hectares distributed over three municipalities that previously constituted an Area of Socioeconomic Influence. These are the municipalities of Agaete (39.2%; 126.04 p/km$^2$), Artenara (43.3%; 18.77 p/km$^2$), and San Nicolás de Tolentino (17.5%; 68.96 p/km$^2$). The unemployment rate is above 30% in the towns of El Risco (0.87%; 3.5 p/km$^2$) and Guayedra (0.17%; 0.23 p/km$^2$). These are the only two localities located in the park. The most representative economic activities are agriculture (corn, tomatoes, papaya, potatoes, aloe, fruit trees), livestock (grazing, two game farms), fishing (traditional and artisanal), rural tourism services, and recreational activities (hiking, adventure). The entire area is considered an Ecological Sensitivity Area because the purpose of protecting the natural park is to preserve the natural resources it contains for public use, education, and scientific research in a manner compatible with conservation. The park, which covers a large part of the northwest of the island, is characterised by the massif formed by the Tirma, Altavista, and Tamadaba mountains (Figure 1) (Appendix B includes a scale topographic government map with all the types of vegetation of the natural protected area (Figure A2). It also incorporates the limits of the municipalities (yellow line) and the limits of the natural area (red line) in correspondence with Figure 1. https://www.idecanarias.es/resources/PLA_ENP_URB/GC/AD/C-09_Tamadaba/152/indice.html (accessed on 2 December 2019)). On the coast, the cliffs of Andén Verde and Faneque stand out. The town of El Risco is considered to be exceptionally representative of traditional agricultural practices. Among the ecosystem services on which its protection is based are the recharging of the aquifer, the protection of soils, facilitating the development of terrestrial and coastal biocenoses, and the maintenance of the biodiversity of the Canary Islands archipelago.

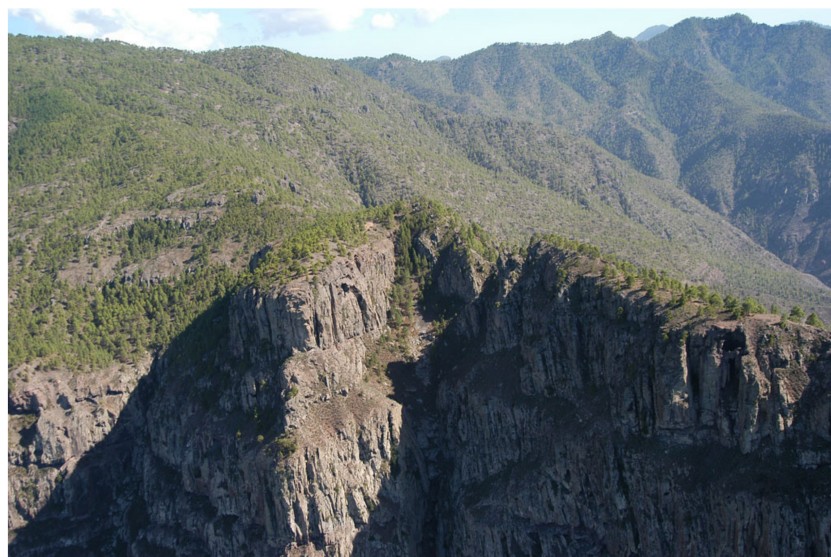

**Figure 1.** Tamadaba Natural Park. Source: Government of the Island of Gran Canaria.

The closest environmental valuation precedent to the study area is the work of León [34]. In this case, the contingent valuation method is applied to obtain the preservation values of the areas of Gran Canaria, which are Cuenca de Tejeda, Cumbres, Tamadaba, and Inagua. The total area to be evaluated amounts to 28,000 hectares. The surveys were carried out by telephone during the last quarter of 1993. Using survival model functions, the following results were obtained: the median of the lognormal function was 4456 pts for 1993. This value multiplied by the adult population of the island of Gran Canaria generates a total social benefit of 2255 million pts (13,522,823 euros in 1993). Table 1 shows the different mean and median values obtained according to the distribution function considered, the median value being more recommendable than the mean, since it has the advantage of not being affected by the extreme values of the sample, also being coherent with the majority rule as a criterion of social welfare in public decision-making.

**Table 1.** Marginal provisions payable (€1999).

| Distribution | Median | Average |
|---|---|---|
| Weibull | 5.19 | 12.54 |
| Lognormal | 4.45 | 16.53 |
| Gamma | 4.01 | - |
| loglogística | 4.50 | 66.92 |
| Exponential | 7.89 | 11.38 |

Source: León [34].

## 3. Method

There is a group of environmental assessment and management methods that are not based on the most orthodox economic analysis of consumer theory [77] and, in recent years, have experienced a high level of use, being applied to deal with different issues. These are known as multi-criteria decision methods [59,63–65,68,71,72,78–80]. They are based on the fact that economic agents do not optimize their decisions on the basis of a single objective, but rather try to reach a compromise solution based on multiple conflicting objectives, or aim to satisfy a series of goals associated with those objectives. Zahedi [81] proved that the alternative selection process is consistent with a respondent's multi-attribute utility function. More recently, the work of Väinö and Ahtiainen [82] proved the relevance of the distribution of weighting weights and their application to inform public policy recommendations in the environmental area.

Although there is abundant literature regarding the use of these methods in the determination of priorities in environmental management in different areas [52,60,61,66,70,71,80,83,84], there is still very little experience with environmental valuation in monetary terms [10,11,67]. This is a bias in the literature that this work contributes to solving. Within the framework of a discrete multi-criteria decision, that is, when the number of alternatives is finite, the analytic hierarchy process (AHP) proposed by Saaty is chosen [85–89]. This method consists of obtaining preferences or weights of importance for the criteria and the alternatives. The AHP allows for a broad understanding of the alternatives to be compared. Using the AHP aims to improve the understanding of how the participants trade-off quantifiable and non-quantifiable attributes of the natural park that determine the total economic value. The application of this method requires that both the criteria and the alternative criteria can be structured in a hierarchical way (Figure 2).

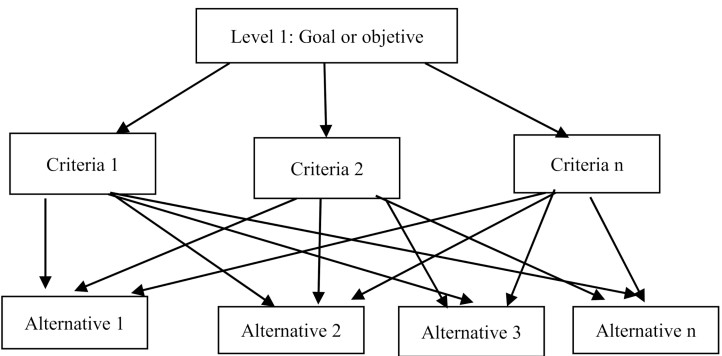

**Figure 2.** Hierarchy for the design of the first and second multi-criteria decision matrix.

The first level of hierarchy corresponds to the general purpose of the problem, the second to the criteria or attributes, and the third to the alternatives. The method proposes to assign a weight vector, $w = [w_1, w_2,..., w_n]$, to the criteria of a certain multi-criteria problem. To do this, comparisons are made of each criterion $i$ with each criterion $j$, obtaining values $a_{ij}$ that can be grouped in a square matrix of order n. This is the so-called binary comparison matrix, $A = [a_{ij}]$. The comparisons through which the decision centre assigns value judgments are made two by two because it is easier for the decision-maker. However, these comparisons would not be reliable if the number n of criteria exceeds the limit of the human brain by $7 \pm 2$ due to a limitation in simultaneous information handling. Saaty justifies the following numerical scale for the estimation of the $a_{ij}$ coefficients (Table 2).

**Table 2.** Saaty rating scale.

| $A_{ij}$ | Valid When Criterion $i$ is Compared with Criterion $j$: |
|---|---|
| 1 | equally important |
| 3 | slightly more important |
| 5 | far more important |
| 7 | arguably more important |
| 9 | absolutely more important |

If criterion $i$ is not equal to or more important than $j$, but the other way around, the value of the coefficient is $1/a_{ij}$. The elements of the diagonal of matrix A are the unit, since each criterion with respect to itself is equally equivalent. The supratriangular part of matrix A is evaluated, that is, the elements of $a_{ij}$ where $j > i$, $\left[\frac{n(n-1)}{2}\right]$.

The rest of the matrix will be made up of the inverse of these coefficients. The A matrices of binary comparisons are of the type of so-called reciprocal matrices, the property on which the efficiency of the AHP method is based. A matrix of binary comparisons is consistent when $a_{ij} = w_i/w_j$, for all $i$, $j$. This means that the relative importance of $i$

versus *j* ($a_{ij}$) is exactly the ratio of its absolute importance, which is its weight ($w_i/w_j$). This property of consistency, although desirable, is not fulfilled because human beings are always somewhat inconsistent, even more so when it comes to finding a compromise between opposing criteria. Therefore, in practice, we must always count on some inconsistency. However, the rate of inconsistency cannot exceed 10%; otherwise, it would be necessary to repeat the questionnaire to make comparisons, at least in this experiment, where only the expert panel surveys are available. The flexibility of the AHP method means that it can be applied individually or by an expert panel [90]. In the latter case, the aggregation of the individual judgements gives the vector of priorities with respect to the alternatives proposed. The procedure to be followed for the aggregation of the individual judgements of the panel of experts will depend on the degree of homogeneity of the judgements. In the case of a homogeneous group of experts, the aggregation of the individual trials, either from the pairs of comparisons or from the vectors specific to each comparison matrix, is carried out by calculating the eigenvector. This is the procedure used here to calculate the total economic value of the Tamadaba Natural Park. When the panel of experts is not homogeneous; that is, when divergent professional groups are integrated into the same panel, integration must be carried out using the procedure known as extended goal programming [68,91].

## 4. Results

The valuation of the Tamadaba Natural Park gives a reference in monetary terms that allows us to make inter-temporal and inter-spatial comparisons for the purposes of its management in the public sphere. The valuation of the space allows us, for example, to quantify the loss of natural capital due to unforeseen events, such as natural disasters or forest fires, or to evaluate investment challenges for mainstreaming resilience into landscapes [92] or to establish priorities for the purpose of allocating public resources for conservation and management. Quantification also allows us to evaluate the necessary compensation or the recovery of natural capital for damages, among many other utilities [93]. Economic valuation of an environmental asset is a reference to the well-being it provides to society, since it is in monetary terms that we usually express our preferences when acquiring a good or service in the market, depending on the utility it provides or the need to be satisfied [94].

Due to the difficulties in applying the most sophisticated methods of assessing environmental quality, both because of the lack of publicly available data and because of financial limitations in obtaining and processing the data, as well as the need to assess each and every one of the components that make up the total economic value, an AHP was chosen to illustrate the potential monetary value of a protected natural area, such as the Tamadaba Natural Park. The total economic value of an environmental asset consists of its direct use value, indirect use value, its option or quasi-option value, existence value, and legacy value, which are defined in the survey used (Appendix A). The aim of the analysis carried out in this paper is to obtain the total economic value of the Tamadaba Natural Park, using Saaty's discrete-choice multi-criteria method. Given that a homogeneous panel of experts was chosen, it was not necessary to apply goal-weighted programming, the eigenvectors themselves being sufficient to obtain an aggregate vector. The economic valuation to be carried out responds to the hierarchy in Figure 3.

For calculation of the direct use value, defined as that obtained directly from the exploitation of the natural resources of the space, the zoning and the specific regime of uses foreseen in the Master Plan for the Use and Management of the Tamabada Natural Park was taken as a reference approved by the Canary Government in 2000 (Figure 4) (Appendix B includes a scale topographic government map with the zoning and categorization of land uses of the natural protected area (Figure A3) in correspondence with Figure 4. https://www.idecanarias.es/resources/PLA_ENP_URB/GC/AD/C-09_Tamadaba/152/indice.html (accessed on 2 December 2019)).

In accordance with this regime of uses, a potential income is estimated, taking into account the provisions of Royal Decree 1492/2011, of 24 October, which approved the

valuation regulations of the Land Law [95] and Royal Decree 7/2015, of 30 October [96], which approved the revised text of the Land and Urban Renewal Law, for the purpose of obtaining a minimum value for the land close to the compensation value, as provided for in the aforementioned legal precepts. The market value of the land, taking into account the regime of uses, is valued by the method of capitalization of rents in accordance with the criteria and rules set forth in the standard, and in accordance with the data on rural leases published by the Ministry of Agriculture, Fisheries and Food of the Government of Spain (https://www.mapa.gob.es/es/estadistica/temas/estadisticas-agrarias/canonesarrendamiento2018_tcm30-523537.pdf (accessed on 2 December 2019)) (Table 3). The direct value thus obtained will be the value on which the remaining values pivot, using the weightings of the panel of experts. This procedure is followed in order to obtain the total economic value of the space, taking into account not only the use values (direct, indirect and option), but also the non-use values (existence and legacy) [9,10,25].

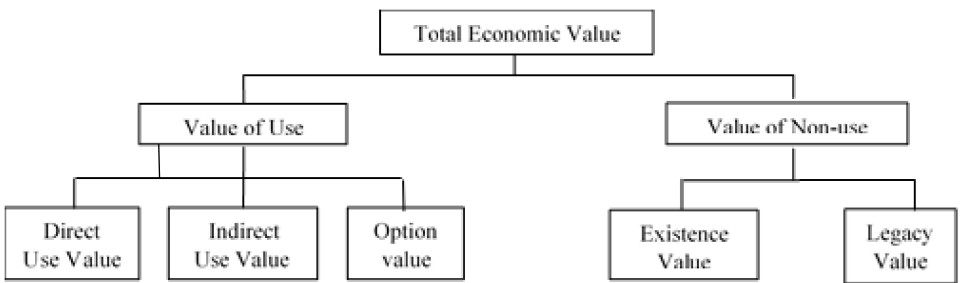

**Figure 3.** Two-level hierarchy for calculation of Total Economic Value.

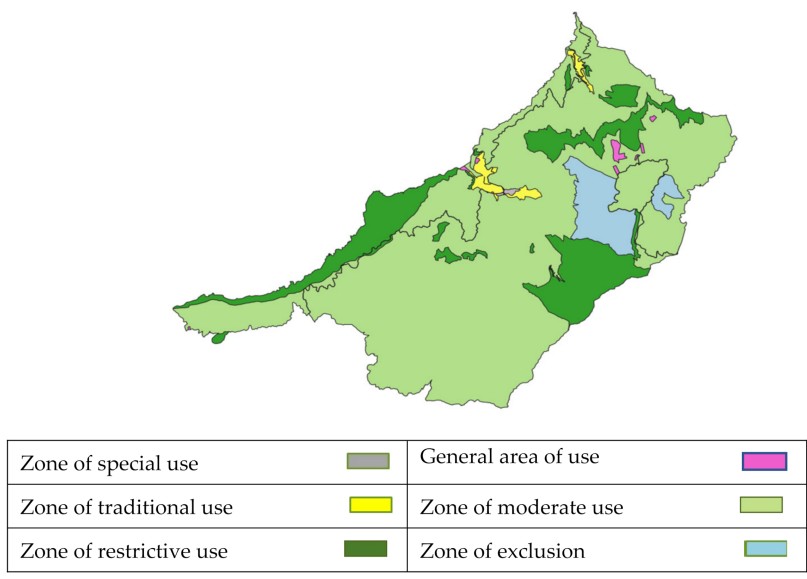

| Zone of special use | | General area of use | |
|---|---|---|---|
| Zone of traditional use | | Zone of moderate use | |
| Zone of restrictive use | | Zone of exclusion | |

**Figure 4.** Tamadaba Natural Park zoning. Source: Government of the Island of Gran Canaria (2020).

In order to obtain the value of the remaining uses (indirect and optional), as well as the values of non-use (legacy and existence), a panel of experts was set up. They assigned, by means of questionnaires (Appendix A), weightings for the purpose of establishing a hierarchy between the different values that make up the total economic value of the natural park, following the procedure foreseen in the methodology described in the previous section. The panel of experts is composed of public officials responsible for the management of the natural park, scientists and technicians. All of them have extensive knowledge of the area as users, managers or researchers. The interviews were conducted in person in some cases, and by telephone in others. The results of the judgments of the panel of experts, already aggregated, are shown in Table 4. To solve the AHP problem, we employ

the R program with the package developed by Glur [97] and illustrated by Cho [98]. The individual preference weights are computed using the dominant eigenvalues method described in Saaty [87].

**Table 3.** Valuation of direct use (Euro 2019).

| Zone Typology | Purpose | Direct Use Value (€/m²) | Total Direct Use Value (€) |
|---|---|---|---|
| Zone of exclusion (397.6 ha.) | Use for scientific or conservation purposes (art.12 PRUG) | | 11,904,529.67 |
| Zone of restricted use (1276.5 ha.) | Reduced public use (art.13 PRUG) | 2.99 | 38,219,648.20 |
| Zone of moderate use (5714.3 ha.) | Public, forestry and agricultural use (art.14 PRUG) | | 171,091,684.87 |
| Zone of traditional use (94.6 ha.) | Traditional agricultural use (art.15 PRUG) | 8.98 | 8,497,246.34 |
| Zone of special use (7.1 ha.) | Pre-existing rural or urban settlements (art.17 PRUG) | 264.43 | 18,777,600.00 |
| General area of use (48.5 ha.) | Use for recreational and educational-environmental activities (art.16 PRUG). | 260.8 | 126,488,000.00 |
| Total Direct Use Value | | | 374,978,709.08 |

Source: Prepared by authors based on the Master Plan for the Use and Management of the Tamabada Natural Park (PRUG), and Spanish Act (2011, 2015).

**Table 4.** Final weightings of the components of the Total Economic Value.

| Weights | Use/Non-Use Matrix | | Direct Use Value/Indirect Use Value/Option Value/Existence Value/Legacy Value—Matrix | | |
|---|---|---|---|---|---|
| | Aggregate Individual Preferences_Eigenvector | Standardized Vector | Aggregated Weights_Eigenvector | Standardized Vector | Final Weighting |
| Direct use value | | | 0.13 | 0.1413 | 0.076 |
| Indirect Use Value | 0.44 | 0.54 | 0.58 | 0.6304 | 0.342 |
| Option Value | | | 0.21 | 0.2282 | 0.123 |
| Existence Value | | | 0.32 | 0.3809 | 0.175 |
| Legacy Value | 0.37 | 0.46 | 0.52 | 0.6190 | 0.284 |
| Total | 0.81 | 1 | | | 1 |

Fifty percent of the expert panel considers that the use value (direct, indirect and option) is more important than the non-use value (existence and legacy). The rest of the panel considers the opposite to be true, as we can see in Figure 1.

From the integration of the individual weights, we obtain the eigenvector shown in Table 4, first column, which is consistent with Figure 5, where the value of use outweighs the value of non-use. More interesting is Figure 6, in which it is possible to observe the aggregated priorities, using a trimmed mean by trimming observations' higher and lower quantiles.

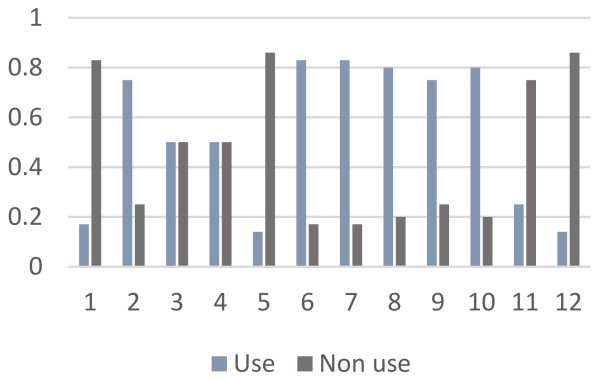

**Figure 5.** Use/non-use individual preference weights.

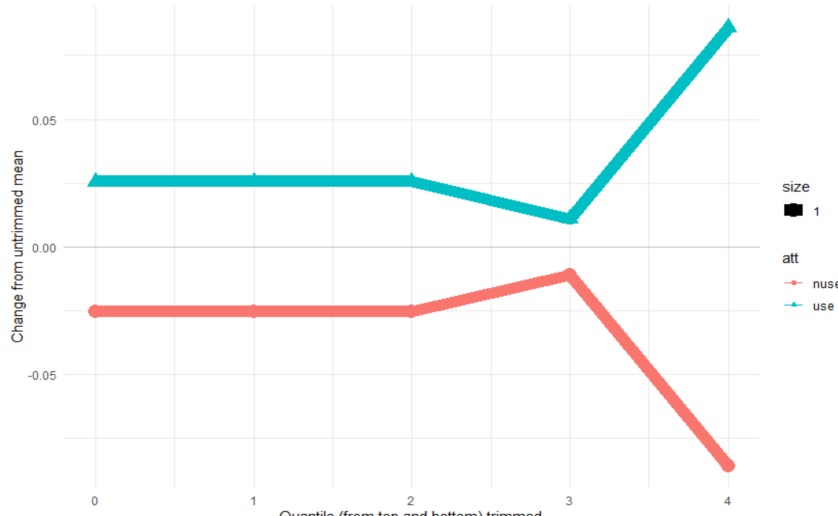

**Figure 6.** Changes of aggregated weights based on quantile of data trimmed (use/non-use values).

The panel of experts converges firstly from the second quartile (median) onwards, whereas in the last quartile they disagree considerably, as shown in Figure 6. Then, discrepancies between the natural park's use/non-use weights increase for the highest weights, but for all quartiles, the use value has more weight than the non-use value. In Figure 7 related to the use value, we can observe that the expert panel assigns greater weight to the indirect use value compared to the direct and option use values for all quartiles. As in Figure 6, in Figure 7, the preference-ordering in the expert panel undergoes a change. The consensus in the expert panel breaks down in terms of the direct use value compared to the option value above the second quartile of the weight distribution, reversing the preference order. In Figure 8, we can observe that the maximum difference between the eigenvalue and mean aggregation is below 0.05 (red line).

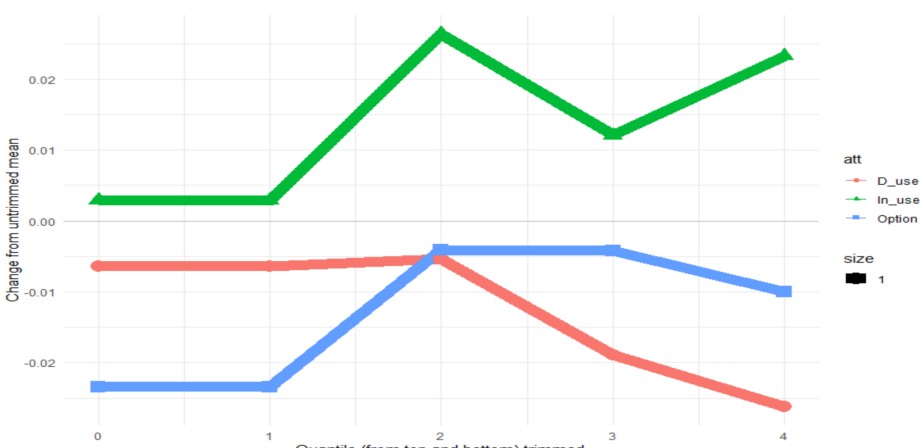

**Figure 7.** Changes of aggregated weights based on quantile of data trimmed (use values).

We observe that the legacy value has more weight than the existence value as non-use values for all quartiles, as shown in Figure 9. There is a clear consensus in the expert panel regarding this preference, as it can be seen in Figure 10 in relation to the individual preference weights for non-use values.. Taking into account, conceptually, each type of value, the result is consistent with the previous figures. Thus, the use value predominates in Figure 6. The highest weight related to the use value corresponds to indirect use, and the weight of the option value is higher than the weight of the direct use value in the third and fourth quartiles (Figure 7). The bequest value is the value assigned to an environmental

good justified by the desire to preserve a given good for the "use and enjoyment" of future generations, which is known in the literature as intertemporal altruism.

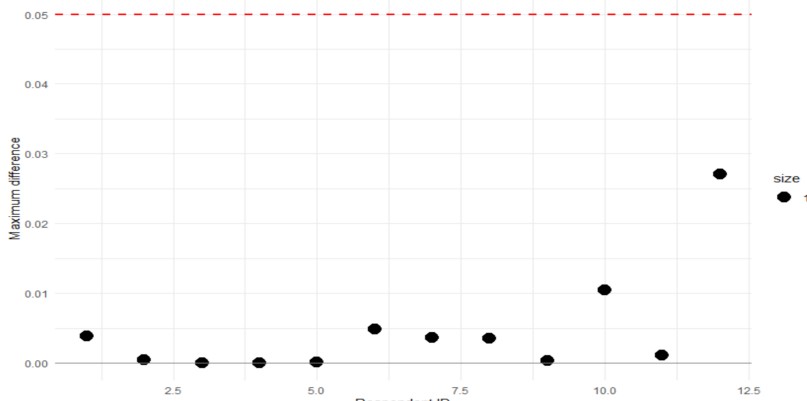

**Figure 8.** Maximum differences between eigenvalue and mean aggregation (use values).

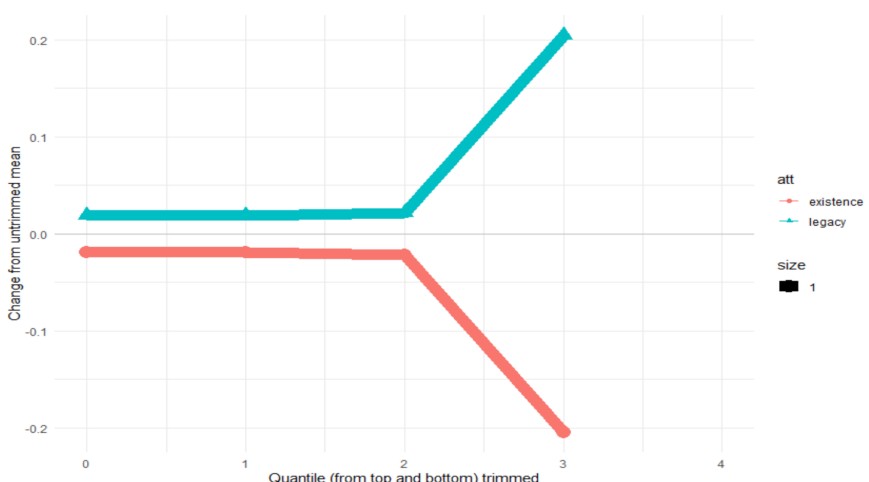

**Figure 9.** Changes of aggregated weights based on quantile of data trimmed (non-use values).

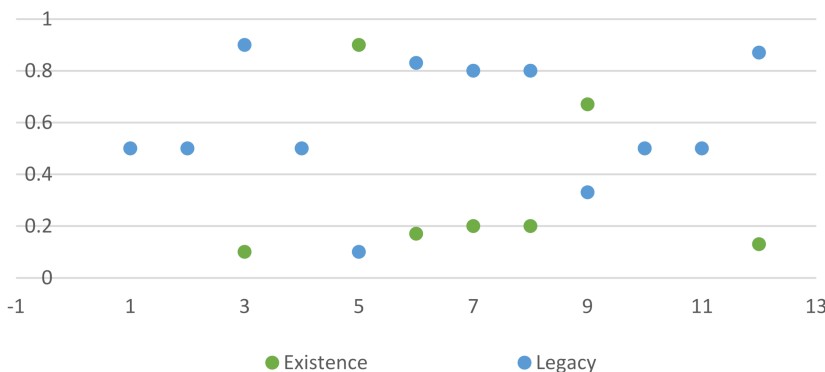

**Figure 10.** Individual preference weights for non-use values.

Their own priority is established by each expert with respect to the different use/non-use values that make up the total economic value. The vectors obtained from the matrix of judgments of each expert are integrated using the dominant eigenvalues method described by Saaty [87], when it is a question of a homogeneous panel of experts for the purpose of obtaining a single prioritisation, as is the case here. This single vector is normalised by

the sum. From this prioritisation vector, and taking the direct use value as a reference, the other values are obtained using the following value ratios [9]:

$$\text{Ratio} = \frac{\sum \text{market value of comparable assets}}{\sum \text{weighting of comparable assets}} \tag{1}$$

$$\text{Unknown asset value} = \text{Ratio} \times \text{ weighting of unknown asset value} \tag{2}$$

The results are shown in Table 5. It is observed that the direct use value of the natural park which comes from the market value only represents 7.6% of the total value, and the remaining values from the non-market represents 92.4% of the total economic value. Taking into account the preferences reflected in the weights assigned (Table 3), the indirect use and legacy values are the highest, representing 62.60% of the total value.

**Table 5.** Total economic value according to its components.

| Type of Value | Final Weighting | Value in Euros | Value per Use (€) | |
|---|---|---|---|---|
| Direct use value | 0.076 | 374,978,709.08 | | |
| Indirect use value | 0.342 | 1,687,404,190.86 | Use | 2,669,256,337.00 |
| Option value | 0.123 | 606,873,437.06 | | |
| Existence value | 0.175 | 863,437,816.96 | Non-use | 2,264,674,045.63 |
| Legacy value | 0.284 | 1,401,236,228.67 | | |
| Total economic value of the Tamadaba Natural Park | | | | 4,933,930,382.63 |

From observation of Tables 3 and 5, it can be concluded that the zones with the greatest restrictions on use, which are those of exclusion, restricted and moderate use, are those with the lowest market value in terms of land valuation, according to the legal criteria for valuation, in the event of compensation for expropriation or property liability derived from public management. However, from an environmental point of view, these are the areas that have the highest weighting because these soils have the highest concentration of indirect use, option, existence and legacy values. This illustrates that this bias in the legal valuation of land sometimes leads to a bias in public decision-making with respect to its management, directing public and private infrastructure toward land with a lower market value because of lower compensation costs. Table 6 shows that the values that generate the greatest heterogeneity in the panel of experts are those corresponding to non-use values (legacy and existence), since they have the greatest standard deviation.

**Table 6.** Heterogeneity among the decision-makers' priorities.

| Stats. | Direct Use Value | Indirect Use Value | Option Value | Existence Value | Legacy Value |
|---|---|---|---|---|---|
| mean | 0.1441 | 0.6007 | 0.2550 | 0.4048 | 0.5951 |
| s.d. | 0.063 | 0.135 | 0.141 | 0.246 | 0.246 |

Thus, the total economic value of the Tamadaba Natural Park is 4933 million euros, with data for the calculation of direct use corresponding to the year 2018 and, updated to euros 2019, is relatively comparable to the 2881 million euros of the total economic value of the L'Albufera Natural Park in Valencia, using data for the calculation of direct use corresponding to the year 2005 [9]. It is necessary to consider the high level of the agricultural lease tax in the Canary Islands with respect to the rest of Spain, as well as the different time period to which the valuations refer. Therefore, we start from a direct use value in terms of a minimum compensation value per hectare that is much higher in the Canary Islands than in the rest of Spain [99,100]. As a research agenda for the future, the value of the area could be disaggregated, taking into account the zonation made in

the planning document of the natural park and its own ecosystems. However, this would require a panel of experts with a higher degree of specialization and knowledge of the environmental and socioeconomic values of each type of zone, as well as its geographic delimitation. This would be necessary to avoid high inconsistency ratios in the value judgments that would invalidate the valuation. These are analyzed in the following section.

## 5. Discussion

*Consistency Analysis*

The consistency indices and consistency ratio give us a measure of consistency of value judgments of the comparison matrix. Saaty [86] showed that when the CR (CR $= \frac{\lambda max - n}{n-1}\left[\frac{1}{RI}\right]$), $\lambda_{max}$ is the maximum eigenvalue of the pairwise comparison vector and $n$ is the number of attributes. *RI* is the random consistency value as a function of the matrix size. The *RI* when three attributes is 0.52 [86]. The ahp.ri and ahp.cr functions in R software (ahp.survey "package") return CR) is higher than 0.1, the choice is deemed to be inconsistent. The Ahp.survey package in the R program allows us to quantify the inconsistency among the decision-makers. In our case, the CR is below 0.05 for each matrix corresponding to each expert, considering them independently. More interesting is observing the boxplots in Figure 11, which allow us to visualize the heterogeneity in the weights that each respondent assigns to each attribute, considering the panel of experts as a whole. Thus, we observe that, in the three matrices related to the value of use, there are two experts of the panel whose CR is higher than the recommended (0.10) (pink point). The rest of the experts are within the recommended CR value. The consistency ratio (CR) for the respondents as a whole in relation to the value of use matrices is acceptable, although there is room for improvement (0.137). As we can see in Figure 7, weights assigned to the indirect use value are higher than the rest of the values or attributes. The smallest dispersion relates to the direct use value which comes from market value, and the highest dispersion relates to the option value, corresponding with the results of Table 6. For land management purposes, it is very interesting to observe where there is heterogeneity in the assignment of the weights in the panel of experts, because this allows us to interact among the experts by conducting a second round of surveys, or by expanding the sample of experts with greater knowledge in those attributes where greater heterogeneity is observed. However, Harker [101] developed a method to replace inconsistent values, using the error matrix [102].

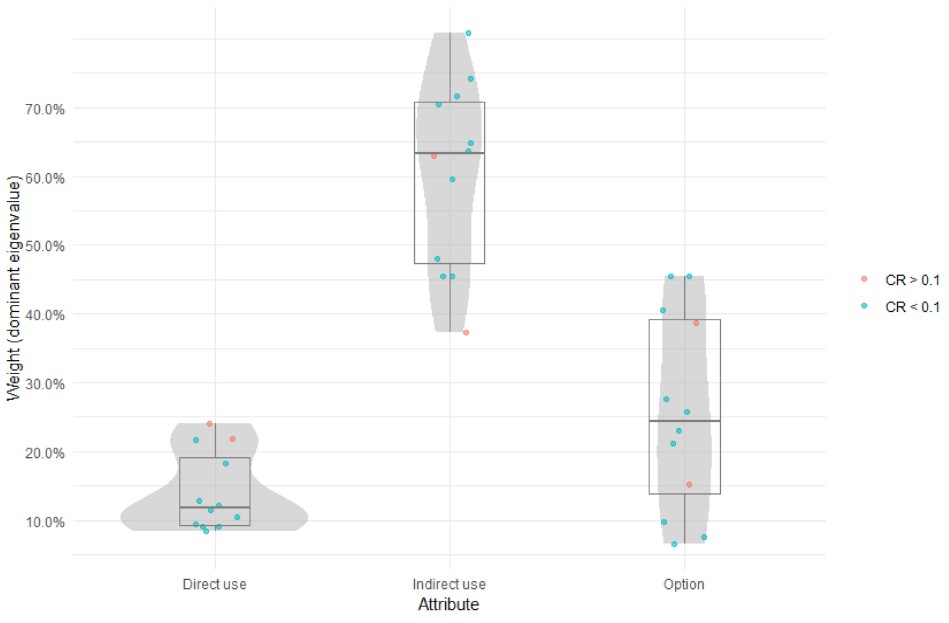

**Figure 11.** Boxplots of individual priorities and consistency ratios.

## 6. Conclusions

Territory management involves the difficult task of determining land activities and uses that are compatible with its degree of protection and conservation. This management is usually carried out within a legal framework, as is the case with protected natural areas. Achieving a balance between the use of the space, its conservation and protection is not an easy task. Rural areas include high ecological value systems, but their vulnerability has not been sufficiently pinpointed in the development indicators. Indeed, it is clear that trying to avoid co-management of natural spaces for better enjoyment and better conservation generates dissatisfaction in those who cannot access, if access limitations are imposed. This social, economic and environmental reality is what drives public decision-making in the field of natural area management, in the search for a balance that maximizes the welfare of all. Even today, a challenge in the area of decision-making for investment in public conservation projects is the preference for time, so that greater priority is given to the future benefit from conservation of natural resources, as opposed to the immediate economic benefit of other types of public projects that generate shorter-term benefits. As explained in the introduction to this article, although Europe has a broad institutional framework for the protection of biodiversity and the ecosystem services provided by natural areas, we detected a bias in relation to the adequate valuation of the multifunctional services provided by rural land and, in particular, land for environmental purposes. This institutional bias triggers an undervaluation that leads to inefficient decisions in the field of public land use management, with consequences for social welfare. Economic valuation of the multifunctional services of forests involves enormous complexity that has been approached from the field of environmental economics and from other disciplines, with the aim of achieving greater efficiency in their management. However, none of these methods is exempt from criticism, either because of the biases that can occur in obtaining the values or because of their limited scope. In any case, they are a useful tool for evaluating the scope that different public investment alternatives can generate in social welfare. Therefore, they allow comparisons to be made between different investment alternatives, providing criteria for their prioritization while taking into account limited budgetary resources.

The analysis carried out in this case allows us to prove the valuation "myopia" from which public decisions are made, such as, on occasion, the location of public and private infrastructure on rural land with lower market value, in terms of direct or productive use. This contributes to lowering the costs of public infrastructure projects due to compensation for expropriation, without taking into account the positive environmental externalities that such land generates. By using the AHP multi-criteria method, using as a reference the market value obtained by applying the rent capitalization method according to the valuation rules and criteria of the Spanish legislation, it is proved that the market value only represents 7.6% of the total economic value of the protected area. Thus, taking only this value as a reference for land management has a high bias, taking into account its effect on social welfare. This problem is even greater in an island context, with a limited territory under enormous urban planning and socioeconomic development pressures. As the Chief of the United Nations, Elliot Harris has already pointed out that if we assign a value to nature, we can quantify it, and if we can quantify it, we can manage it. If we manage its value, we avoid its destruction.

Thus we observe that, in the three matrices related to the value of use, there are two experts on the panel whose CR is higher than the recommended (0.10) (pink point). The rest of the experts are within the recommended CR value. The consistency ratio (CR) for the respondents as a whole, in relation to the value of use matrices, is acceptable, although there is room for improvement (0.137). As we can see in Figure 7, weights assigned to the indirect use value are higher than the rest of the values or attributes. The smallest dispersion relates to the direct use value which comes from the market value, and the highest dispersion relates to the option value, corresponding with the results of Table 6. For land management purposes, it is very interesting to observe where there is heterogeneity in the assignment of weights by the panel of experts, because this allows us to interact

among the experts by conducting a second round of surveys, or by expanding the sample of experts with greater knowledge in those attributes where greater heterogeneity is observed. However, as pointed out in the previous section, a method to replace inconsistent values is developed, using the error matrix. At a later stage, it would be desirable to extend the detailed analysis of the ecosystem services provided by the nature park, within the framework of each type of area and use, distributing the monetary values spatially. This would provide us with very useful information for the purposes of spatial management and decision-making in order to minimise the environmental impact of certain actions and to make the conservation of certain areas compatible with the direct income provided by the park. The economic value of the functions and services provided by the natural park, according to which areas would decrease well-being in the event of their loss, would also allow us to prioritise investment according to the economic value of the functions and services for the areas of the natural park.

**Author Contributions:** Conceptualization, A.H.-A.; methodology, A.H.-A.; validation, J.C.S.; investigation, N.C.-P.; resources, J.C.S.; data curation, A.H.-A.; visualization, N.C.-P. All authors have read and agreed to the published version of the manuscript.

**Funding:** This research was partially supported by the European Union's Horizon 2020 research and innovation program under grant agreement 101037424, project ARSINOE (Climate resilient regions through systemic solutions and innovations).

**Informed Consent Statement:** Informed consent was obtained from all subjects involved in the study.

**Data Availability Statement:** The data presented in this study are available on request from the corresponding author.

**Acknowledgments:** The authors would like to thank the panel of experts for their dedication to this experiment, without whose participation and knowledge it would not have been possible to complete it. Special mention and thanks are due to the Director of the Master Plan for the Use and Management of the Tamabada Natural Park (PRUG) of the Government of the Canary Islands.

**Conflicts of Interest:** The authors declare no conflict of interest.

## Appendix A. Survey

*Appendix A.1. Introduction*

Understanding value as the capacity of a good or service to satisfy a need or provide well-being or delight, all the values that make up an environmental good, will be defined below. It should be clear that the economic value to be obtained should in no case be identified with the price, which results from the free confluence between suppliers and demanders of goods and services. This scenario does not occur in the case of environmental goods because they lack an explicit market for exchange. The TOTAL ECONOMIC VALUE of an environmental good is composed of USE VALUES and NON-USE VALUES.

1. Value of Use (V.U.): It is the value of the goods and services obtained by the people who use the environmental asset as an instrument; 1.1. either by directly exploiting the natural resources (agricultural cultivation, livestock exploitation, forestry exploitation, etc.), obtaining a market value or price from it, giving rise to what is known as direct use value (D.U.V.); 1.2. either by using the space indirectly through the provision of services that have no market value or price ($CO_2$ fixation, nutrient retention, soil retention, aquifer recharge, support to other ecosystems, enjoyment of recreational areas, etc.), giving rise to what is known as indirect use value (I.U.U.V.); 1.3. I.U.); 1.3. either because the potential future use of the natural space is unknown (medicinal use of flora and fauna, etc.) and its loss produces in the person a loss of well-being, and therefore assigns a value to these uncertain or probable potential future uses. This is known as option value or quasi-option value (O.V.). Therefore, the use value is composed of the direct use value plus the indirect use value plus the option value (value of not closing the possibility of a future use of the good) or quasi-option

value. (It is a somewhat more complex concept than option value, but both have the uncertainty that, in this case, has to do with the irreversibility of the decision maker's erroneous decisions, either due to lack of scientific knowledge regarding the impact of the decision or due to lack of information).

2. Value of Non-Use (V.N.U.): It is the value that environmental assets have for certain people, which has nothing to do with the use of the space. It is a value that is not linked to use. The two fundamental attributes of this value are: existence value (E.V.) and legacy value (L.V.). The existence value refers to the value assigned by certain people who are not users of the space and do not intend to be but who value positively the fact that the property exists. Its disappearance would cause a loss of welfare even if they do not use it or do not plan to do so in the future. Related to this concept would be, for example, use for research purposes. Preserving an environment, an ecosystem, a species makes it possible to preserve a laboratory for experimentation and research. Bequest value is the value assigned by people to an environmental good justified by the desire to preserve a given good for the enjoyment of future generations (intertemporal altruism).

Thus, the values that we are trying to obtain correspond to the following scheme:

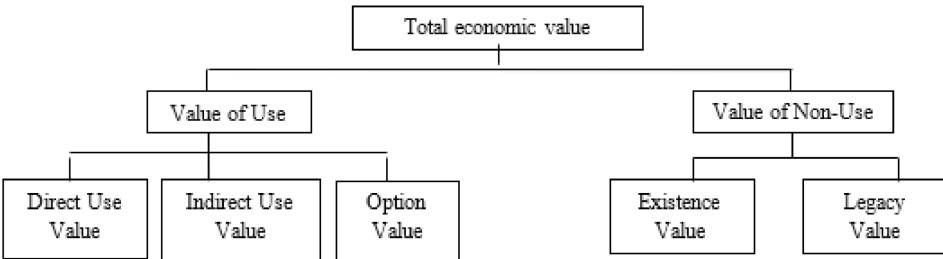

**Figure A1.** Diagram of the environmental, economic and social services of Tamadaba Natural Park that integrate the above values, configuring its total economic value.

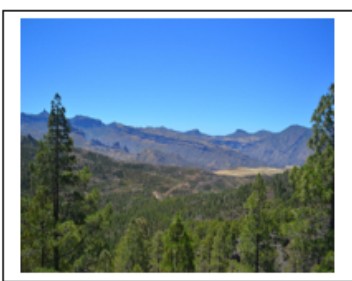
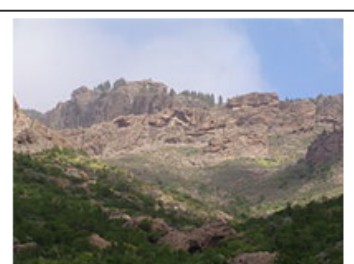
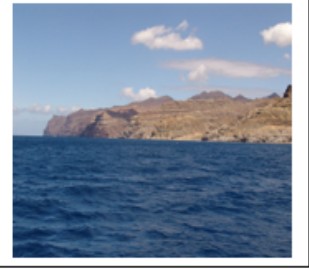

Photo: Iván Hernández Cazorla          Photo: Bergwelt          Photo: Rüdiger Marmulla

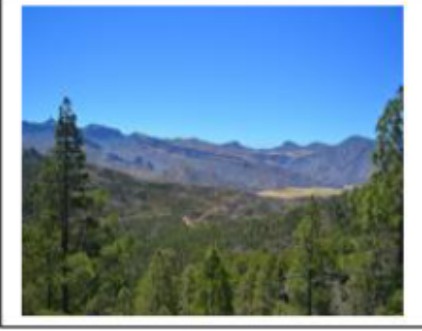
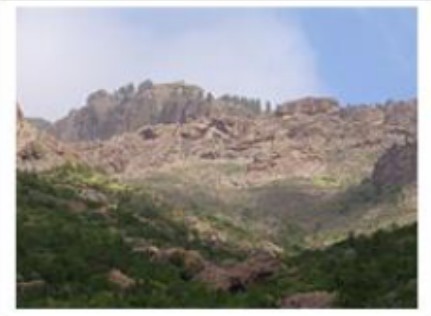
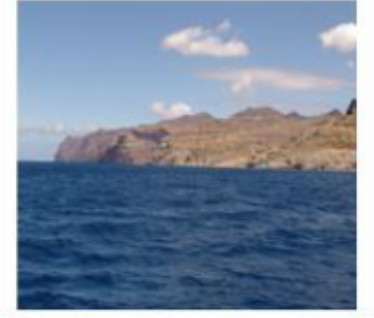

Photo: Iván Hernández Cazorla          Photo: Bergwelt          Photo: Rüdiger

| |
|---|
| *Province:* Las Palmas; ***Date of creation:*** 1987 *N° **Locations:*** Agaete, Artenara and la Aldea de San Nicolás. |
| Its greatest singularity is the network of ravines, escarpments and massifs of this area that form an erosive landscape of contrasts, where natural elements of notable geomorphological interest can be identified. Tamadaba is home to one of the best preserved natural pine forests on the island, with a remarkable efficiency in hydrological catchment. Other well-preserved biosystems are those of the ravines and the cardonales and tabaibales of the lowlands, as well as the escarpment habitats. The Guayedra cliffs are home to several endangered species, some of which are exclusive to this area. In addition, certain endemic and endangered bird species find ideal nesting areas in the pine forests. From a cultural point of view, there are also some isolated and semi-abandoned farmhouses, and the archaeological importance of areas such as El Risco and the Guayedra valley. The entire area, with the exception of a sector in the town of El Risco, was considered an area of ecological sensitivity. This area has been declared a special protection zone for birds5 for the Conservation of Wild Birds.1 The coastal sector, from La Aldea to Agaete, is considered a point of geological interest because it allows "visualizing the external part of the Tejeda caldera, with areas of hydrothermal alteration (Los Quemados) and samples of the sillic facies that overflowed the caldera (El Risco)". Within the territorial scope of the natural park there is a great variety of habitats in which plant communities of notable ecological value and more than 200 plant species develop. Up to 16 of these habitats are considered priority habitats by the Habitats Directive and include a strong endemic component and the presence of endangered species. Among the endemic species, there are six species that are exclusive to the park, most of them rock-dwelling species, 33 endemic species of Gran Canaria and 64 of the Canary Islands archipelago. The fauna is equally remarkable, especially birds and reptiles. Among the birds, two jewels of the Canarian avifauna stand out, such as the Gran Canarian woodpecker and the Gran Canarian blue chaffinch. In the whole area of the park, but especially in the lower areas, the Gran Canarian giant lizard can be observed, as well as the green or blue-tailed mullet. The invertebrates, much more numerous but more difficult to perceive, present a high degree of endemism, from those widely represented as the Gran Canarian grasshopper to the endemic beetle, exclusive of the fossil beach of Punta de las Arenas. Wherever there are coastal beaches, especially in Estancia del Manso and Punta de las Arenas, they are formed by mobile dunes. |

| Summary of environmental and socioeconomic services and products of Tamadaba NP | |
|---|---|
| Environmental and socioeconomic attributes Environmental and commercial products | Environmental and socioeconomic attributes Environmental and commercial products |
| Natural habitats representative of the ecological systems of the Canary Islands: natural pine forest, cardonales-tabaibales, escarpment habitats. | $CO_2$ retention |
| Aquatic ecosystems | Aquifer recharge |
| Exclusive species of flora and fauna | Climate change protection |
| Endemic vertebrates and invertebrates | Water purification |
| Unique avifauna | Biomass exploitation |
| Traditional settlements | Agricultural, livestock and forestry resources |
| Handicrafts | Cultural services |
| Leisure and recreational activities | Natural heritage |
| Traditional agricultural, livestock, hunting and hydrological activities. | Biological diversity |
| Natural landscapes | Mineral resources, water resources |
| Presence of geomorphological structures and singular formations. | Tourism activities: hiking, guided tours … |
| Paleontological sites | Local products |
| Hydrological catchment | Research and educational services |
| Natural pine forest | Fishing exploitation |
| Archaeological importance: Risco Caído | Recreational uses: camping area, beaches, … |
| Archaeological importance: Guayedra Valley | |

**Q.0:** Self-assessment: From 0 (=not at all) to 5 (=maximum), what is the degree of knowledge of Tamadaba N.P. that you consider you have? ……….. (put the number).

*Appendix A.2. Valuation Methodology*

Of the various existing multi-criteria methods, we will use Saaty's Analytic Hierarchy Process (AHP) proposed in the early 1970s (Saaty, 1972) and further developed. This method is flexible and allows multiple applications. In this case, it will be used to order a valuation problem. For the establishment of priorities, judgments are made by means of pairs of comparisons, that is, the elements are compared in pairs from the point of view of the criterion. The comparison matrix of the pairs of judgments is filled in with numbers representing the relative importance of one element with respect to another under certain characteristics.

| Grade of Importance | Definition | Explanation |
|---|---|---|
| 1 | Equal importance | Two activities that contribute equally to the objective. |
| 3 | Moderate importance | The experience and judgment of one activity is slightly in favor of the other. |
| 5 | Strong importance | The experience and judgment of one activity are strongly favored over the other. |
| 7 | Very high importance | One activity is favored over the other very strongly; its dominance is demonstrated in practice. |
| 9 | Extreme importance | The favorable evidence of one activity over the other is of the highest possible order of affirmation. |
| 2, 4, 6, 8 | For trade-offs between the above values | Sometimes it is necessary to interpolate a numerical judgment of commitment because there are no good figures to describe it. |
| 1, 1-1, 9 | For linked activities | When the elements are very close and almost indistinguishable, if the difference is minimal (1.1) and if it is maximal, within moderation, (1.9). |

### SURVEY
Name and surname: … … … … … … … … … … … … … … … … …… … … … … … … …
Academic Background: …. … … … … … … … … … … … … … … ..… … … … … … … …
Occupation: … … … … … … … … …… … … … … … … … … … … … … … ...… … …
(Note: the data will be treated anonymously)

### A. Matrix Type 1 (2 × 2)_Criteria:

P.1.: According to Saaty's scale, and in relation to Tamadaba NP, which criterion is more important in determining the total economic value: the use value or the non-use value? And how much more important is it, according to Saaty's scale? Put the grade in row 1, column 2.

| | Value of Use | Value of Non-Use |
|---|---|---|
| Value of Use | 1 | |
| Value of Non-Use | | 1 |

### B. Matrix_Type 2 (3 × 3)_Attributes_Criteria 1:

P.2.1: According to the Saaty scale, in relation to Tamadaba NP, and taking into account the use value, which attribute is more important in determining the use value—the direct or indirect use value? And how much more important is it, according to Saaty's scale? Put the grade in row 1, column 2.

P.2.2: According to the Saaty scale, in relation to Tamadaba NP, and taking into account the use value, which attribute is more important for the purpose of determining the use value—the direct use value or the option value? And how much more important is it according to Saaty's scale? Put the grade in row 1, column 3.

P.2.3: According to the Saaty scale, in relation to Tamadaba NP, and taking into account the use value, which attribute is more important for the purpose of determining the use value—the indirect or option use value? And how much more important is it according to Saaty's scale? Put the grade in row 2, column 3.

| | Direct Use Value | Indirect Use Value | Option or Quasi-Option Value |
|---|---|---|---|
| Direct use value | 1 | | |
| Indirect use value | | 1 | |
| Option or quasi-option value | | | 1 |

### C. Matrix_Type 3 (2 × 2)_Attributes_Criterion 2

P.3.: According to the Saaty scale, in relation to Tamadaba NP, and taking into account the NON-use value, which attribute of Tamadaba NP is more important in determining the

NON-use value—the existence value or the bequest value? And how much more important is it according to Saaty's scale? Put the grade in row 1, column 2.

|  | Existence Value | Bequest Value |
|---|---|---|
| Existence value | 1 |  |
| Bequest value |  | 1 |

**Appendix B. Map of the Natural Park of Tamadaba**

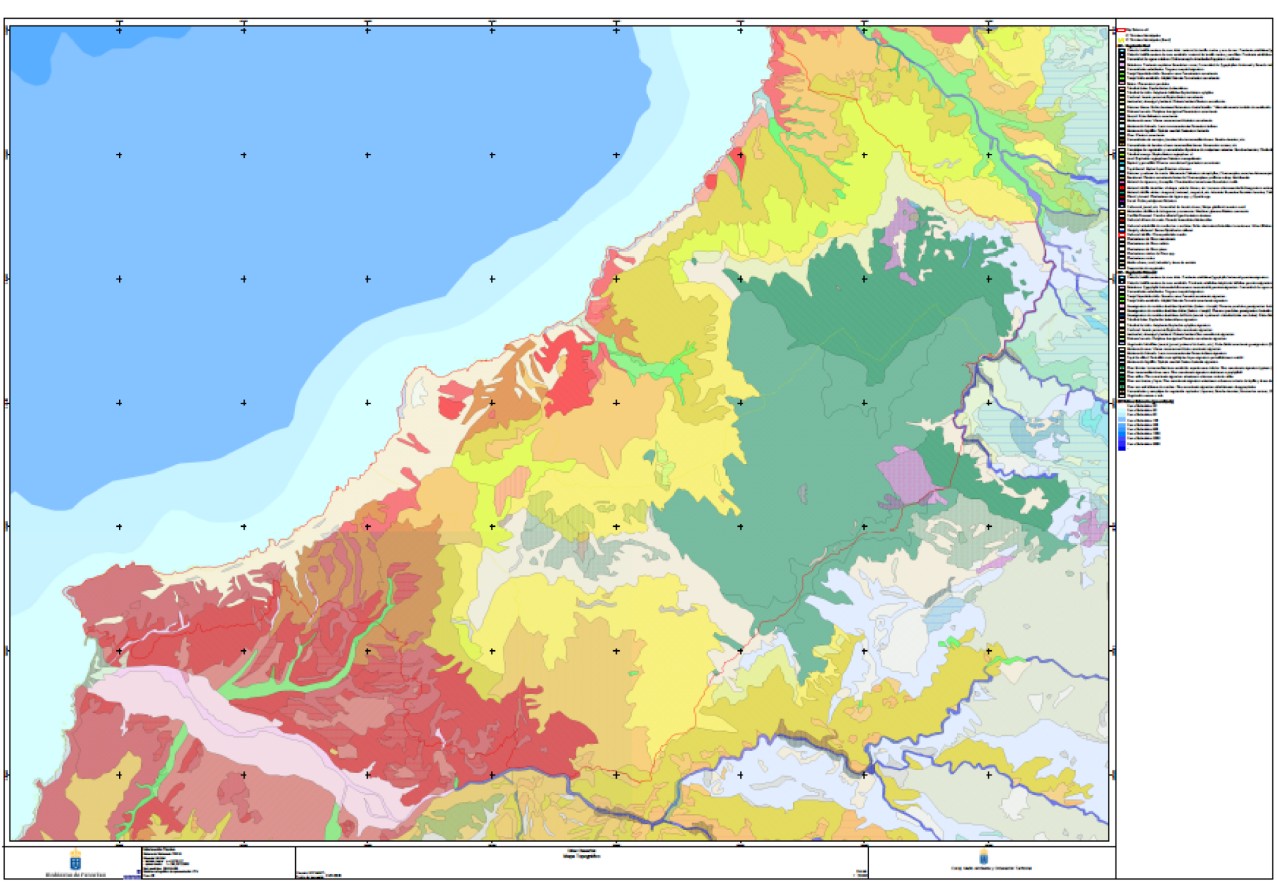

**Figure A2.** Topographic map of the area of Tamadaba, Gran Canaria. Source: Government of the island of Gran Canaria.

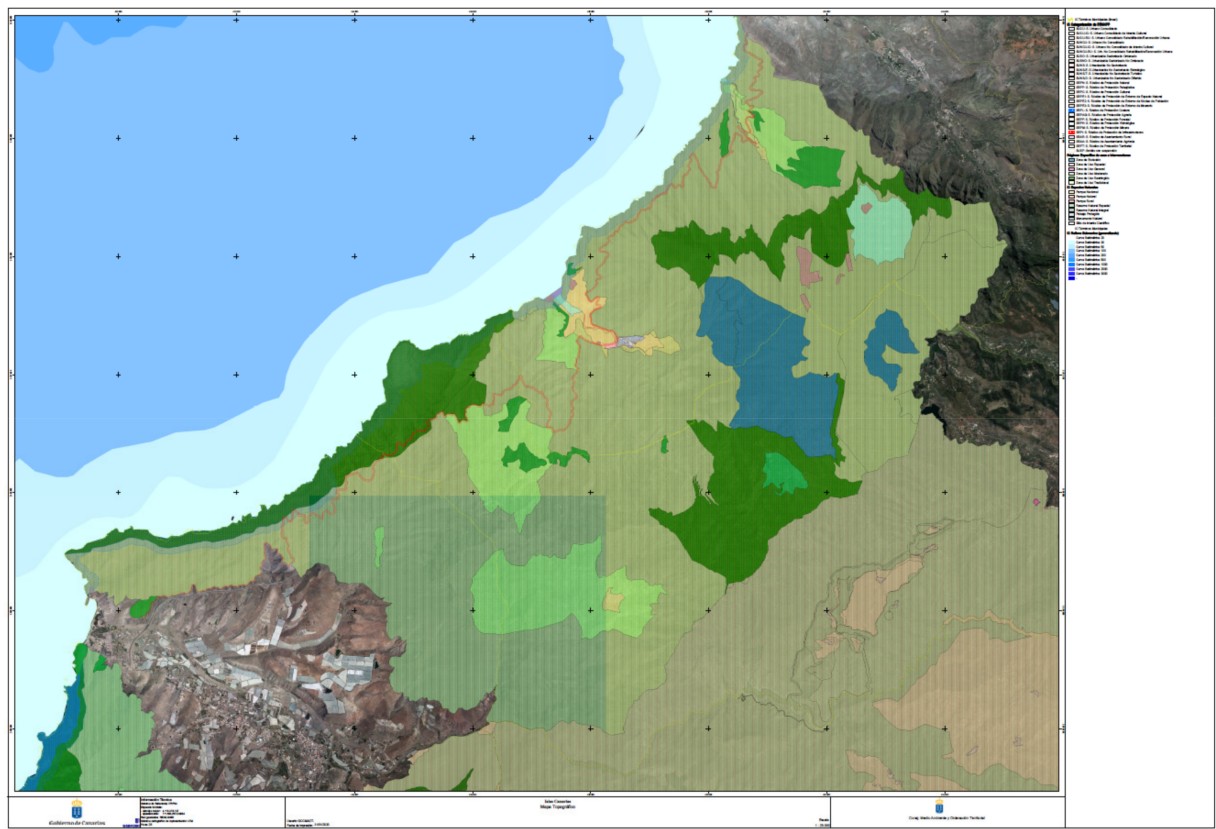

**Figure A3.** Map of land use in the area of Tamadaba, Gran Canaria. Source: Government of the island of Gran Canaria.

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
