# Peer review of "Rethinking Legal Criteria for Assessing Compensation for Rural Land Expropriation: Towards a European Institutional Framework"

_land, doi:10.3390/land11020194_

Round 1
Reviewer 1 Report
In introduction - terms "existence" and "legacy", "option values" needs to be clarified
Figure 1 and 4. - are not clear enough
Author Response
We would like to start acknowledging anonymous reviewer #1 for the efforts made in this review. We believe that the constructive comments made have resulted in the improvement of the text.
Specific comments:
- In introduction - terms "existence" and "legacy", "option values" needs to be clarified
For a better understanding of the different types of values considered in the analysis, the survey used for the case study describes each one. Then, we have considered to include the survey as Annex I.
- Figure 1 and 4. - are not clear enough
We have changed the Figure 1. This new Figure includes scale and coordinates. Moreover, we add two maps of the natural park of Tamadaba with the geographical characteristics in the annex II (Figure A1 and Figure A2). These are scale topographic government maps which include all the types of vegetation of the natural protected area; the limits of the municipalities; the zone typology, and the categorization of land uses in correspondence with Figures 1 and 4. The Figure 4 and Figure A2 in the annex II are similar. We refer too to more information it is possible to find in: https://www.idecanarias.es/resources/PLA_ENP_URB/GC/AD/C09_Tamadaba/152/indice.html

Reviewer 2 Report
I found the paper interesting and sound. I would see it published in the journal. However, the present version should be amended and, in particular, I urge authors to better document and justify the use of a specific case study. Justification and generalization is required to help readers interpreting your results in a broader perspective. Language usage should be also simplified (shorter sentences,...).
Author Response
We would like to start acknowledging anonymous reviewer #2 for the efforts made in this review. We believe that the constructive comments made have resulted in the improvement of the text.
Comments to the Author
I found the paper interesting and sound. I would see it published in the journal. However, the present version should be amended and, in particular, I urge authors to better document and justify the use of a specific case study. Justification and generalization is required to help readers interpreting your results in a broader perspective. Language usage should be also simplified (shorter sentences, ...).
We have added a paragraph to justify the use of a specific case study and the need of generalizing the results (Line 55):
“,… The reflection is supported in a specific case study as we can see in the previous literature, using qualitative methods [1, 30, 34-35, 42, 45-46]. However, the aim of this work is not so much to illustrate a specific case study, but to make evident the need of an institutional framework that could be a support for land valuation; in particular, for rural or environmental uses [44]. The reflection involves governance in a wide sense (policies, institutions, process to make decisions, valuation, etc). European institutional framework provides a wide range of legal tools to protect biodiversity [2, 4]; however, there is a lack relate to the valuation of rural or environmental land. Not a few European projects need land for their development, some of which get into a conflict with the sustainable uses one. That is why this reflection should be transferred to a broader legal framework than the Spanish one, although it serves as an example [49, 51-53, 55-56].”

Reviewer 3 Report
Dear Authors,
Thank you for sending your paper to journal Land. Your paper is interesting and in the focus of the journal. You are tackling the legal criteria for assessing compensation for land expropriation and using the analytic hierarchy process for land management with multifunctional services. This is a quite challenging process for rebuilding or building in old spaces.
Before your paper goes to the next stage of reviewing process, I want to point out you some improvements in the current version of the paper:
- I am not a native speaker, but there are some text errors. Please recheck for grammatical and spelling errors.
- The title is too long. Hard to understand.
- In your paper, you are using quite of lot so-called chain citations, for example: [1-6]. Each citation deserves two to three sentences about why it is important for your research.
- Page 2 line 69 please use a standard form for formulas. Do not incorporate in text. Also, the meaning of the formula is important.
- Footnotes are very long. Think purpose of introducing the footnotes in the paper. Legal text usually has footnotes so long.
- The title of chapter 2 has “previous work”. This need to be part of the literature review.
- “The surveys were carried out by telephone during the last quarter of 1993.” Completely outdated information. It is almost 30 years old! Not relevant anymore!
- Data from table 1 are from 1996! See my previous comment.
- In chapter Method, you have a matrix of citation. Please read my comment number 3.
- “a panel of experts was set up to assign, by means of questionnaires, weightings for the purpose of establishing a hierarchy between the different values that make up the total economic value of the natural park” – this is the most important part for starting AHP. So, it would help if you had an appendix questionnaire/survey. Then a detailed analysis of experts.
- AHP is a well known and used method, so please do not skip steps.
- All numbers in tables align to the right; the same variable used some numbers of decimal places. In that way, it is easy to compare numbers.
- In table 4 you have different weights. What is the meaning of each of the weights. For example UDV?
- Instead of Graph use Figure.
- Do not start sentences with on figure, in table, …
- Graph 2 to 5 are of poor quality. Hard to read it.
- Page 11 line 302 ??. – surplus
- The conclusion can not have any citations.
The part of your paper regarding the AHP is not straightforward. As I point out, AHP is a well known and used method. Used the knowledge of others scholars.
Regards,
Author Response
We would like to start acknowledging anonymous reviewer #3 for the efforts made in this review. We believe that the constructive comments made have resulted in the improvement of the text.
Comments to the Author
Dear Authors,
Thank you for sending your paper to journal Land. Your paper is interesting and in the focus of the journal. You are tackling the legal criteria for assessing compensation for land expropriation and using the analytic hierarchy process for land management with multifunctional services. This is a quite challenging process for rebuilding or building in old spaces.
Specific comments:
Before your paper goes to the next stage of reviewing process, I want to point out you some improvements in the current version of the paper:
- I am not a native speaker, but there are some text errors. Please recheck for grammatical and spelling errors
We have sent it to a native speaker.
- The title is too long. Hard to understand
We have reduced it in a shorter and generalized than the previous one. This is the new title:
“Rethinking legal criteria for assessing compensation for rural land expropriation. Towards a European institutional framework”.
- In your paper, you are using quite of lot so-called chain citations, for example: [1-6]. Each citation deserves two to three sentences about why it is important for your research
We have reviewed the chain citations and explain their contribution to the literature on this topic.
- Page 2 line 69 please use a standard form for formulas. Do not incorporate in text. Also, the meaning of the formula is important
We have changed the form in text by formula. We explain from line 73 onwards that this formula is a correction factor that contemplates the articles 7.3 and 17.5 of the Royal Decree 1492/2011.
- Footnotes are very long. Think purpose of introducing the footnotes in the paper. Legal text usually has footnotes so long
The longest footnote is # 1 in relation to the legal text. We have decided to delete it because its details with many legal nuances do not have a greater effect on the discussion. Moreover, we have removed the footnotes from 4 to 8 relate to values definition because we add an annex I with the survey that explain these types of values.
- The title of chapter 2 has “previous work”. This need to be part of the literature review
We have included the literature of this chapter in the chapter before. Thus, we eliminate the expression “previous work” in this chapter.
- “The surveys were carried out by telephone during the last quarter of 1993.” Completely outdated information. It is almost 30 years old! Not relevant anymore!
In this case, the date of the work referred to is not the most relevant thing, which is specified for comparative purposes from the quantitative point of view if appropriate and with due caution by the analyst. What determines the reference to this work is that it is the only study carried out in the field of environmental valuation coinciding with part of the area studied in this work. The determining factor in this comparison is the destination of the land being compared, which in both cases is natural, ecosystemic or environmental. Thus, it would be a pity not to reference it, because although it is extemporaneous, it allows us to appreciate the importance of adequately valuing the land in all its characteristics and functions, because even using different methodology and being extemporaneous, the comparison allows us to illustrate the bias between the market value exclusively, and the social and environmental value of the target area. Undoubtedly, it is a source of inspiration.
- Data from table 1 are from 1996! See my previous comment
In our opinion, this question is answered in the previous section.
- In chapter Method, you have a matrix of citation. Please read my comment number 3
We have reviewed the chain citations and explain their contribution to the literature on this topic.
- “a panel of experts was set up to assign, by means of questionnaires, weightings for the purpose of establishing a hierarchy between the different values that make up the total economic value of the natural park” – this is the most important part for starting AHP. So, it would help if you had an appendix questionnaire/survey. Then a detailed analysis of experts
We have added an annex with the survey used for the case study for a better understanding of the analysis (annex I). The composition of the panel of experts, and the form of conducting, is specified in lines [266-267]. However, given that we are dealing with a small panel of experts with the objective of illustrating the potential of the methodology in this case, we understand that what is most relevant is to analyze the discrepancies in the panel of experts that support the assigned weights in order to illustrate the desired consensus. If it were a broad panel of participants of different composition, or if they formed a social matrix (citizens, public decision-makers, and experts), a broad analysis of the panel of experts could be justified, and even the self-weighting of their knowledge could be used. In this case, not only is the panel of experts small, but it is also quite homogeneous both in terms of subject matter knowledge and spatial knowledge.
- AHP is a well known and used method, so please do not skip steps
Taking into account the extensive theoretical and empirical literature on the AHP methodology, we understand that it is easier to understand the reflection and reading with the exposition made. It is true that we do not explain all steps to obtain the results in Table 4 because you can check the procedure in the references Glur [97] and Cho F. [98].
- All numbers in tables align to the right; the same variable used some numbers of decimal places. In that way, it is easy to compare numbers
We have changed it in this way.
- In table 4 you have different weights. What is the meaning of each of the weights. For example UDV?
We have eliminated the abbreviations.
- Instead of Graph use Figure
We have replaced Graph by Figure.
- Do not start sentences with on figure, in table, …
We have reviewed the paper to change the sentences in this way.
- Graph 2 to 5 are of poor quality. Hard to read it
We have replaced all graphics from 2 to 5 to a larger size. However, we have certain limitations in this sense related to the AHP package in R program.
- Page 11 line 302 ??. – surplus
We have removed them.
- The conclusion can not have any citations
We have moved these references to the appropriate section.
- The part of your paper regarding the AHP is not straightforward. As I point out, AHP is a well known and used method. Used the knowledge of other scholars
The purpose of this paper is to illustrate, through reflection and the application of the AHP methodology to a specific case, the distortion in the valuation of rural land as a result of the legal obligation to observe a methodology that corresponds to the 19th century, in order to obtain the "most probable market value" of a rural land as a compensatory value (method of capitalization of rents). This is not only at odds with scientific progress in valuation, since, as illustrated in the specific case, multiple environmental services that are not and cannot be the object of transactions are left out of the valuation; but also, and as a consequence of this undervaluation, biased decisions are made regarding public management of land use. Thus, the article aims to illustrate, on the one hand, the limitations derived from the institutional sphere that affect the public management of land uses with obvious consequences for social welfare; and on the other hand, on a broader and more generic level, to demonstrate the need for science, and its appropriate application to the specific case, to illustrate decision-making in the public sphere. In this sense, an effort is made in the new wording to highlight the need for a legal framework at the European level to support decisions on land use that guarantee its sustainability.

Reviewer 4 Report
Dear authors,
This article deals with the very interesting and increasingly important issue of land valuation. Although it is strongly rooted in Spanish reality, the problem itself as well as the algorithm of its solution are of universal character. The obtained result of the very high non-market value of land is consistent with the results of studies in other countries. I have a few comments, more of editorial nature.
Firstly, the title, in my opinion, is unnecessarily long and unnecessarily starts and refers to legal criteria, which are not the main subject of the analysis and of this article. I suggest shortening it.
Secondly, the map presenting the study area (Figure 1) does not meet the basic criteria of a map - it is hardly legible, there is no scale or legend, it is difficult to read the borders of the area, nor is it possible to find the places mentioned in the text. It is not clear where the area from Figure 4 is located on this map? I suggest replacing it with a larger and more readable map.
Thirdly, the panel of experts is the key source of parameters for estimating the non-market value, so it should be better described in terms of the number of experts, the way they were recruited and the areas of expertise or institutions they represented, as well as the form of conducting the panel itself.
Fourthly, in table 4 in the first column (weights) abbreviations were used, which even if you can guess what they mean, as formally they were not explained anywhere in the text - they should be explained somewhere or referred to earlier in the text (they are already given in table 5 (also 6) - they should be developed in the same way in table 4).
Fifthly, the explanations in Graph 2 and Graph 3, Graph 4, Graph 7 should be given in a bigger font as they are illegible.
Finally, I believe that the conclusions should refer to the universality of the presented procedure and emphasise the possibility of its use also in other geographical and institutional contexts.
Congratulations on a very interesting article.
Author Response
We would like to start acknowledging anonymous reviewer #4 for the efforts made in this review. We believe that the constructive comments made have resulted in the improvement of the text.
Comments to the Author
Dear authors,
This article deals with the very interesting and increasingly important issue of land valuation. Although it is strongly rooted in Spanish reality, the problem itself as well as the algorithm of its solution are of universal character. The obtained result of the very high non-market value of land is consistent with the results of studies in other countries. I have a few comments, more of editorial nature.
Specific comments:
- Firstly, the title, in my opinion, is unnecessarily long and unnecessarily starts and refers to legal criteria, which are not the main subject of the analysis and of this article. I suggest shortening it.
We have reduced it in a shorter and generalized than the previous one.
- Secondly, the map presenting the study area (Figure 1) does not meet the basic criteria of a map - it is hardly legible, there is no scale or legend, it is difficult to read the borders of the area, nor is it possible to find the places mentioned in the text. It is not clear where the area from Figure 4 is located on this map? I suggest replacing it with a larger and more readable map.
We have changed the Figure 1. This new Figure includes scale and coordinates. Moreover, we add two maps of the natural park of Tamadaba with the geographical characteristics in the annex II (Figure A1 and Figure A2). These are scale topographic government maps which include all the types of vegetation of the natural protected area; the limits of the municipalities; the zone typology, and the categorization of land uses in correspondence with Figures 1 and 4. The Figure 4 and Figure A2 in the annex II are similar. We refer too to more information it is possible to find in:
https://www.idecanarias.es/resources/PLA_ENP_URB/GC/AD/C09_Tamadaba/152/indice.html
- Thirdly, the panel of experts is the key source of parameters for estimating the non-market value, so it should be better described in terms of the number of experts, the way they were recruited and the areas of expertise or institutions they represented, as well as the form of conducting the panel itself.
We have added an annex with the survey used for the case study for a better understanding of the analysis (annex I). The composition of the panel of experts, and the form of conducting, is specified in lines [266-267].
- Fourthly, in table 4 in the first column (weights) abbreviations were used, which even if you can guess what they mean, as formally they were not explained anywhere in the text - they should be explained somewhere or referred to earlier in the text (they are already given in table 5 (also 6) - they should be developed in the same way in table 4).
We have eliminated the abbreviations from the Table 4. Lines from 268 to 275 explain the way to obtain the results which can be seen in Table 4. It is true that we do not explain all steps to obtain it because you can review it in the references Glur [97] and Cho F. [98] to avoid hindering an agile reading. However, we have added a specific reference to the methodoly explained in the section before, and added the survey in the Annex I to better understanding.
- Fifthly, the explanations in Graph 2 and Graph 3, Graph 4, Graph 7 should be given in a bigger font as they are illegible.
We have replaced all graphics from 2 to 5 to a larger size. However, we have certain limitations in this sense related to the AHP package in R program. Graph 7 has not been changed because it is important to appreciate all dispersión of the points to better understanding the coherence of the answers. If we increase the size of the graph some points mix with others, so we can not apprecite the position of each other.
- Finally, I believe that the conclusions should refer to the universality of the presented procedure and emphasise the possibility of its use also in other geographical and institutional contexts.
The introduction and conclusions sections justify the need for a European legal framework to support public management of land use, particularly in relation to rural and environmental land. Beyond the specific case presented in this paper, the underlying reflection is the possibility that the protection of rural and environmental land be included in the European agenda with an adequate assessment of each and every one of its characteristics and multi-functions to avoid biases in public decision-making in relation to land use.

Round 2
Reviewer 3 Report
Dear Authors,
Thank you for the new version of your paper.
Regards,